# On the Influence of Religious Assumptions in Statistical Methods Used in Science

**Cornelius Hunter**

Natural Sciences, William Jessup University, Rocklin, CA 95765, USA; chunter@jessup.edu

**Abstract:** For several centuries, statistical testing has been used to support evolutionary theories. Given the diverse origins and applications of these tests, it is remarkable how consistent they are. One common theme among these tests is that they appear to be founded on the logical fallacy of a false dichotomy. Is this true? It would be somewhat surprising if such diverse and historically important works are all guilty of the same naïve fallacy. Here, I explore these works and their historical context. I demonstrate that they are not logically fallacious, but instead incorporate and require a religious assumption about how a Creator would act. I conclude that this religious assumption and its influence on science should be considered in models of the interaction between science and religion.

**Keywords:** evolution; null hypothesis significance testing; NHST; *p*-value

---

## 1. Introduction

A recent scientific study (Baum et al. 2016) used a series of computational experiments to quantify the evidential support for common ancestry among the primate species. Hundreds of different species were studied, using a total of 28 experiments. The experiments used statistical analyses involving the so-called *p*-value. The *p*-value is a metric of the probability of the experimental observations if the so-called null hypothesis is true. In this case, the null hypothesis was a random design (used to model separate ancestry). If the *p*-value is sufficiently small (because of non-random patterns between the species) then, according to this method, the null hypothesis (separate ancestry) is said to be rejected and the alternative hypothesis (common ancestry) is accepted. Hence, the *p*-value was used as a gauge for how well common ancestry was supported by the experimental evidence. The smaller the *p*-value, the stronger the evidence for common ancestry. In the study, the *p*-values were exceedingly small. For example, one of the experiments yielded a *p*-value of $10^{-1791}$. Such values are much smaller than is often found in scientific studies, to the point that the authors simply reported a value of zero for several of the experiments. The authors reported that they had "found tremendously strong support for the CA [common ancestry] of all primates", and that common ancestry "is an overwhelmingly well-supported hypothesis" (Baum et al. 2016, pp. 1354–55).

Although this study amassed an impressive array of data and computational power, it was by no means a new or unique approach. The study was more of a capstone effort on top of a long history of previous, similar origins studies using *p*-values and null hypothesis testing. The overall approach is referred to as Null Hypothesis Significance Testing (NHST), as explained in Section 2. Those earlier studies, reviewed in Section 3, also provided very powerful evidence in their own right for common ancestry. Indeed, a textbook made the following triumphant conclusion based on just one of the previous studies from 1982:

> Penny et al. suggested that this pattern is so compelling that the [null] hypothesis of separate ancestry should be emphatically rejected as being incompatible with the data. While science rarely deals with certainty, it is fair to say that evolution from common ancestry is now supported beyond any reasonable doubt. (Baum and Smith 2012, p. 23)

Beyond these contemporary studies of evolution and common ancestry, this general method has a long history, as reviewed in Section 4. For instance, it also was used in the eighteenth century to support various theories of cosmic evolution, although this work predates our modern statistical theory and nomenclature.

Taken at face value, it constitutes powerful evidence and a power argument for both biological and cosmological evolution, but it comes at a cost. As Section 5 explains, applications of NHST to confirm biological and cosmological evolution entail an underlying religious assumption, namely, that the Creator would create in a random manner consistent with a full creation. The potential parallel with plenitude thinking is beyond the scope of this paper. This religious assumption often is unspoken and unrecognized. As such, this use of NHST in origins studies provides an excellent case study of how science and religion can interact in ways not typically understood. In these instances, religion exerts a strong, overarching influence on science. Far from the problematic warfare thesis, which envisions religion and science in conflict, here religious assumptions influence and direct scientific conclusions. This paper reviews this example of NHST, the underlying religious content, and its implications.

## 2. General Problems with NHST

The NHST method determines whether the null hypothesis can be rejected. To do this, it computes the $p$-value, which is a metric of the probability of the observed data under the null hypothesis. If the $p$-value is sufficiently small, then the null hypothesis is rejected. Here, we encounter the first problem with NHST: the $p$-value does not actually supply the probability of the observed data under the null hypothesis. The problem is that it is not possible to compute such a probability for continuous variables. So instead, the $p$-value provides a sort of stand-in value. The $p$-value is the probability of obtaining a test statistic at least as extreme as the observed test statistic.

Nonetheless, a low $p$-value tells us something. If the observed data are unlikely under the null hypothesis, then this suggests the alternative hypothesis is true, or at least more likely to be true. This much is clear, but precisely how low must the $p$-value be in order to gain such confidence?

It is tempting to think of the $p$-value simply as the probability of the null hypothesis. In this view, if the $p$-value is found to be 0.01, then the null hypothesis has a probability of 1%, which leaves 99% as the probability of the alternative hypothesis. While this view has often been the implicit message of research papers, it is false. A favorite threshold, provided by Ronald Fisher in his early work, is 0.05 (Fisher 1925). Using this threshold, if your $p$-value falls below 0.05, then the evidence for the alternative hypothesis is assumed to be strong.

However, this is not true. In fact, if you simply repeated an experiment a few dozen times, even if the null hypothesis was true, you likely would obtain a $p$-value below 0.05 in at least one of the experiment trials. This has led to so-called *p-hacking*—the practice of gaming experiments to obtain a sufficiently low $p$-value (Nuzzo 2014).

I will stop here, even though there are more problems with NHST and its $p$-value. There is now a long history of scientific studies with impressive $p$-values that turned out to be unrepeatable and useless. There is also a long history of scientific papers expressing concern about this state of affairs in science. Consider Jacob Cohen's paper with the humorous title, "The Earth is Round ($p < 0.05$)". Cohen bemoans not merely the fact that NHST is problematic but, even more so, that such problems have been known and largely have been ignored for so long. Cohen writes, "After 4 decades of severe criticism, the ritual of null hypothesis significance testing—mechanical dichotomous decisions around a sacred 0.05 criterion—still persists" (Cohen 1994). Largely as a consequence of the $p$-value problems, scientific research has been called into question, (Ioannidis 2005) and some scientific journals have banned the use of $p$-values altogether. For many, NHST and the $p$-value have done more harm to science than good. The problem has become so acute that the American Statistical Association (ASA) took the unprecedented step of issuing a policy statement on a statistical practice, namely, the use of $p$-values. That statement concluded that "By itself, a $p$-value does not provide a good measure of evidence regarding a model or hypothesis" (Wasserstein and Lazar 2016, p. 132). In a later editorial, the ASA

explained that the policy statement "was developed primarily because after decades, warnings about the don'ts had gone mostly unheeded" (Wasserstein et al. 2016, p. 1).

Against this backdrop of longstanding problems and controversy surrounding NHST and *p*-values, its use in origins studies would seem to be suspect. There is, however, another problem with the use of NHST and *p*-values in origins studies. This problem is much more serious.

## 3. Use of NHST in Origins Studies

Consider two example uses of NHST. First, a drug manufacturer develops a therapy for a disease and uses NHST to evaluate the outcomes of test subjects. Were their outcomes significantly different than what could be expected from mere chance (with no therapy)? The null hypothesis, which the manufacturer hopes can be rejected, states that any improvements observed are merely due to random chance. In the tests, most of the sample test subjects experience reduction in the disease symptoms far beyond what could be expected from chance. The *p*-value is extremely small, such that the null hypothesis is rejected, and the therapy is declared a success, supported beyond any reasonable doubt.

Second, an origins researcher develops an evolutionary hypothesis and uses NHST to evaluate it. The evolutionary theory predicts that the species form a common descent pattern, represented by an evolutionary tree. Note that while evolution normally is held with extremely high confidence and presented as a scientific fact, such confidence is temporarily withheld as the purpose of the test is to help to establish evolution's high claims. The null hypothesis states that the species are unrelated, such that similarities found are due merely to random chance. So, the null hypothesis is random design and the alternative hypothesis is common descent by evolution. Research performed on a large sample of species shows many similarities, far beyond what could be expected from the null hypothesis random-chance model. The *p*-value is very small, such that the null hypothesis is rejected, and the evolutionary hypothesis is declared to be true, supported beyond any reasonable doubt.

These two examples share several similarities. Perhaps most importantly, both are pattern-based. That is, both of them expect to see a certain pattern in the test results, and both use, as a null hypothesis, the random chance case in which there are no patterns. However, these two examples also have some important differences. In the therapy example, the experiment is controlled, and the two hypotheses are, by definition, logically complementary. One must be true and the other must be false. Furthermore, the two hypotheses are well understood. For the alternative hypothesis, the researchers were present for the manufacture and testing of the therapy, and they have observed test subjects who were given the therapy. For the null hypothesis, the researchers observed test subjects who were not given the therapy. In other words, the researchers have data for both the alternative and null hypothesis conditions, allowing them to perform the test.

None of this is true in the evolutionary hypothesis example. The test is not controlled, and the two hypotheses are not well understood. Unlike the therapy example, for the alternative hypothesis, researchers were not present over the deep time of evolution to observe common descent. Furthermore, researchers do not know how the evolution of the species occurred. For example, we have observed small-scale adaptations, but whether these extrapolate to the larger-scale change evolution requires remains controversial even among evolutionists (Erwin 2000; Erwin 2017; Jablonski 2017). In addition to the alternative hypothesis, the null hypothesis is also not well understood, for researchers have never observed such a world. The null hypothesis calls for the species' designs to be randomly arranged with no patterns. This is a purely hypothetical world with no correspondence to reality. The only real-world example we have is our world, in which the species' designs are significantly correlated. Indeed, it is not even clear that such a random biological world is even possible. Species exist and interact within ecological niches. These interactions are crucial in biology, and they include nutrient, energy, and information flow between species. It is not clear how this would work, or if it could work, in a randomly arranged world.

So, unlike the therapy example in which the two hypotheses are well understood, in the evolutionary hypothesis example, the null and alternative hypotheses are not well understood. This presents

significant problems with hypothesis testing in origins studies. For example, given these uncertainties, not only do we not know how the species arose, we do not even know the set of all possible ways that the species could have arisen. In this way, the evolution example is quite different than the disease therapy example. In the therapy example, a rejection of the null hypothesis implies the alternative hypothesis is true. They are logically complementary, but this is not so in the evolution example. In the evolution example, the null hypothesis is random design and the alternative hypothesis is common descent by evolution. But it is not known that common ancestry is the only alternative to random species. This means that there could be other explanations for the observed patterns amongst the species. In fact, a rejection of the null hypothesis simply means that the observed data are not random. Otherwise, it says little about the actual structure of the data, and how well (or badly) they fit the evolutionary common descent model. This is an important distinction. In NHST, the null and alternative hypotheses must be carefully designed such that they are logically complementary. Therefore, if the null hypothesis is false, then the alternative hypothesis must be true, by definition. However, in the evolution example this does not hold. Not only does the failure of the null hypothesis say nothing about the veracity of the alternative hypothesis, but a small *p*-value can be obtained even if the data have a low probability on the alternative hypothesis. In other words, simply put, the data might not fit both the null and alternative hypotheses very well.

Clearly, there are significant challenges in this evolutionary hypothesis example. However, this is not merely an abstract example. This evolution example reflects actual studies that have been published. As described above, the studies use a random pattern null hypothesis test, show that the observed species data do not fit the model well, reject the null hypothesis, and argue forcefully for evolution, the alternative hypothesis. I summarize them here in the remainder of this section.

In an influential 1982 study, researchers used five different proteins (cytochrome C, hemoglobin A, hemoglobin B, fibrinopeptide A, and fibrinopeptide B) to infer the evolutionary relationships among eleven different species (rhesus monkey, sheep, horse, kangaroo, mouse, rabbit, dog, pig, human, cow, and ape). The proteins vary somewhat between the species and can be used to infer which species are more closely related, and which are more distantly related. Under the common descent alternative hypothesis, each protein should produce similar evolutionary relationships. On the other hand, according to the researcher's random pattern null hypothesis, the protein relationships between species are random. That is, the evolutionary relationships inferred from the different proteins should not be similar, beyond what might be expected by chance.

In the results, the different proteins did lead to some significantly different evolutionary relationships. For example, one protein placed the dog relatively far from the human (9 species distant out of a possible 10) whereas other proteins show the dog relatively close to the human (3 species distant out of 10). The same was true for the mouse. Nonetheless, there were many similarities in the evolutionary relationships inferred from the different proteins. The evolutionary relationships were certainly not random, between the different proteins, and the null hypothesis was easily rejected. The paper claimed this provided "strong support" for the theory of evolution (Penny et al. 1982, p. 200). As noted above in the Introduction section, one textbook referring to this research concluded, "While science rarely deals with certainty, it is fair to say that evolution from common ancestry is now supported beyond any reasonable doubt" (Baum and Smith 2012, p. 23). This study was also discussed at length in an evolution textbook in a chapter explaining the evidence for evolution (Ridley 1993, pp. 50–54).

In a 2013 study, researchers used extant homologous protein sequences in different species to infer sequences of hypothetical ancestral proteins. The researchers found that the ancestral sequences in related groups of species converged, which they argued was unlikely on their random chance null hypothesis. They concluded that "the probability that chance could produce the observed levels of ancestral convergence ... is $\approx 1 \cdot 10^{-132}$" (White et al. 2013, p. 1). Such an extremely low *p*-value gave the researchers great confidence in their results. They concluded, "Given our results, we suggest that researchers need to be more assertive that evolution has both occurred, and continues to occur" (White et al. 2013, p. 8).

In the large-scale 2016 study discussed in the Introduction section above, researchers used 28 different computational experiments to quantify the evidential support for common ancestry among the primate species. These various experiments typically used a random chance null hypothesis that the observed data did not fit very well, producing extremely low *p*-values. As with the other examples, this gave the researchers great confidence in the veracity of evolution. They concluded that they had "found tremendously strong support for the CA [common ancestry] of all primates," and that common ancestry "is an overwhelmingly well-supported hypothesis" (Baum et al. 2016, pp. 1354–55).

These examples evaluated an evolutionary common descent hypothesis by comparing patterns observed in biological data with a hypothetical random chance null hypothesis. As explained above, neither the alternative hypothesis (evolutionary common descent) or the null hypothesis (random pattern) were known to be feasible. This makes the rejection of the null hypothesis dubious. Furthermore, the evolutionary common descent hypothesis is not known to be the only alternative to the null hypothesis, adding further questions to the meaning of rejecting the null hypothesis. For instance, one study of large-scale genetic data showed that a competing non-evolutionary model performed much better than the common descent model (Ewert 2018). These challenges, both logical and empirical, undermine the researchers' claims that these NHST studies confirm evolution and common descent. I next examine the historical context of these claims, and what they actually mean.

## 4. NHST in Its Historical Context

The previous section discusses studies that used statistical hypothesis testing to make triumphant claims about the evolution of the species. These claims amount to the following reasoning: either the species are randomly arranged, or evolution is true. This reasoning and the conclusion appear to be entirely without merit. This section steps back and reviews the history of such arguments.

In the eighteenth-century, prior to the rise of biological evolutionary theory, a quantitative, statistical hypothesis testing approach was used as a confirmation for cosmological evolution (Bernoulli [1734] 2009; Buffon [1749] 1781; Kant [1755] 2008; Brush 1996). This predates modern statistical theory and the formalization of NHST and the *p*-value, but otherwise this approach parallels the biological studies reviewed above. In my discussion below, for convenience I will use modern terminology to describe these eighteenth-century studies, taking care not to impute ideas not already there. For a more detailed description, see (Hunter 2019).

In these cosmological studies, the form of the argument was that the orbital characteristics of the planets in the solar system revealed striking non-random patterns. Similarities amongst the planets and their satellites included the inclination angle of their orbits about the sun, their direction of travel in those orbits, the direction of their spin about their axes, and the direction of the orbits of their satellites. In these studies, the null hypothesis (again, to use modern terminology) stated that the orbits of the planets and their satellites were randomly distributed across the full range of possible values. For example, rather than the planets' inclination angles being close to zero (with reference to the ecliptic), on the null hypothesis they are randomly distributed across the full range of 0–180 degrees. The striking consistency of these orbits was highly unlikely on this random pattern null hypothesis. Bernoulli considered the inclination angle alone. He gave three different calculations, all of which showed that the odds against the null hypothesis were astronomical. He picked the middle result of the three, which was that the odds of such a coincidental alignment are 1,419,856 to 1. Bernoulli determined this value by dividing the range of possible orbital inclination angles into 17 bins. The orbital inclination angles of the six known planets all fell into the same bin. If the planetary orbits were inclined at random angles, the chances of them all falling into the same bin would be 1 in $17^5$, or 1 in 1,419,856. Given such a low probability, Bernoulli rejected the null hypothesis and concluded his alternative hypothesis must be true, that the orbits were formed by the sun's atmosphere. He who would deny this, concluded Bernoulli, "must reject all the truths, which we know by induction" (Bernoulli [1734] 2009).

Immanuel Kant used similar reasoning twenty years later in his ambitious and expansive cosmological treatise. Kant's random pattern argument now added ten moons in addition to the six

planets, and included additional orbital parameters: the axial rotation, and revolution about the sun, in addition to the orbital inclination angle. In each case, the null hypothesis called for the orbits to fill the full range of possibilities. With these additional parameters, the null hypothesis became even more improbable. Kant claimed this argument provided "proof that the collective movements arose and were determined in a mechanical way in accordance with general natural laws", and that we can entertain "no doubts about it" (Kant [1755] 2008, p. 117). This was proof that the solar system arose mechanistically in accordance with general natural laws.

Similarly, Buffon found that "By the doctrine of chances" the odds of such a coincidental alignment of the planetary inclination angles would be 7,692,624 to 1 (Buffon [1749] 1781, p. 65), and concluded that "It is therefore extremely probable, that the planets were originally parts of the sun" (Buffon [1749] 1781, p. 80). Laplace replaced Buffon's idea with his Nebular Hypothesis. The Nebular Hypothesis called for a cloud of material about the sun that rotates and condenses to form the planets and sun. It was, claimed Laplace, the "true system of the world" (quoted in Brush 1996, p. 22). Laplace made several calculations, eventually finding the odds of the solar system's patterns to be 537 million to 1 if they had arisen by chance (Brush 1996, p. 21).

These eighteenth-century cosmological hypothesis tests were of the same form as the later biological tests. They referred to characteristics of the bodies in the cosmos rather than characteristics of the species, but otherwise closely parallel the later biological tests. Specifically, they were pattern-based, describing the null and alternative hypotheses in terms of patterns that should or should not be observed. All of these cosmological hypotheses used as a null hypothesis the random chance case where the observed data should distribute across the full range of possible values. Beyond this, neither the random pattern null hypothesis nor the alternative hypothesis was well understood. Indeed, for the null hypothesis, today's cosmology holds that such a randomly structured solar system may not be stable (Teyssandier 2018, pers. comm.). For the alternative hypotheses, the specific mechanisms offered for the evolution of the solar system did not explain the observed variations, such as in the inclination angles. In other words, the consistency of the planetary orbit inclination angles (they are all relatively close to the ecliptic plane) was used as powerful evidence against the null hypothesis, but nonetheless variations in those inclination angles could not be explained by the theories. Finally, as with the later biological tests, although the null and alternative hypotheses were not known to be the only possible explanations in these cosmological tests, they nonetheless made triumphant claims upon rejecting the null hypothesis. So again, we see that the reasoning amounts to either the structure of the solar system is random, or the practioners' particular mechanistic hypothesis must be true. As with the biological tests, the conclusions are entirely unwarranted. Or are they?

## 5. The Religious Assumption in NHST Origins Applications

In origins studies, the testing of evolutionary models against random chance models has a long, important history, involving influential scholars. Today, it remains a powerful and important argument for biological evolution. However, as we have seen above, these arguments have several problems, not the least of which is that they appear to be based on a false dichotomy. Simply put, either the world was created to look random or it evolved. Could it be that this important tradition, involving influential thinkers over a three-century span, amounts to nothing more than a logical fallacy? This section argues that there is no logical fallacy but, instead, there is a crucial religious assumption at work.

The testing of evolutionary models against random chance models, in an NHST mode of argument, can be saved from the fallacy of a false dichotomy if the null and alternative hypothesis are, in fact, logically complementary. As discussed in the above sections, this cannot be known to be true from a strictly empirical perspective. However, this could be a starting point within a religious context. Consider a theological position that holds that if the world was created, then it would manifest random patterns. When such patterns are not observed, they would therefore be concluded to have arisen by some sort of evolutionary process. This is how the argument has traditionally been formulated.

For example, Kant claimed that rejecting the null hypothesis (using modern terminology) provided proof of his evolutionary theory. To elucidate on the proof, Kant focused on the question of why the planets revolve about the sun in the same direction, for "it is clear that here there is no reason why the celestial bodies must organize their orbits precisely in one single direction, unless the mechanics of their development had determined the matter" (Kant [1755] 2008, p. 118). If they were arranged by the "immediate hand of God" then we would expect them to reveal deviations and differences.

> Thus, God's choice would not have the slightest motive for tying them to one single arrangement, but would reveal itself with a greater freedom in all sorts of deviations and differences. (Kant [1755] 2008, p. 118)

Here, Kant is making a theological claim, involving a divine intent for the fullness of creation. Overtones of the principle of plenitude (Lovejoy 1936) are obvious but actual influences or connections, if there are any, are beyond the scope of this paper. The point merely is that Kant's argument is beholden to a religious assumption. The planets reveal a pattern rather than a full arrangement with "all sorts of deviations and differences". Therefore, the planetary orbits must have originated not by the immediate hand of God but by natural processes. Kant was therefore certain the solar system arose from a condensing cloud of particles.

The theological content is also evident in the later biological tests. Whereas Kant found similarities between the planets to be powerful evidence for cosmological evolution, Darwin found similarities between the species to be powerful evidence for biological evolution. There should be no such obvious pattern; instead, these anatomical designs should fill the design space. Just as Kant called for "all sorts of deviations and differences" amongst the planets under the design hypothesis, so too Darwin called for more variation amongst the species:

> We never find, for instance, the bones of the arm and forearm, or of the thigh and leg, transposed. Hence the same names can be given to the homologous bones in widely different animals. (Darwin 1859, p. 434)

As with the shared inclination angles of the planetary orbits, these shared features across the species could not be explained on the creation model, for "On the ordinary view of the independent creation of each being, we can only say that so it is;—that it has so pleased the Creator to construct each animal and plant" (Darwin 1859, p. 435). As with Kant, theology supplied the crucial premise in Darwin's argument, allowing him to claim the consistent pattern of arm and forearm, and thigh and leg, observed in the species, as strong evidence for his theory of biological evolution.

After Darwin this religious tradition continued. Here is how Mark Ridley explains this biological test, updated to include the universal genetic code:

> Homologous similarities between species provide the most widespread class of evidence that living and fossil species have evolved from a common ancestor. The anatomy, biochemistry, and embryonic development of each species contains innumerable characters like the pentadactyl limb and the genetic code: characters that are similar between species, but would not be if the species had independent origins. (Ridley 1993, pp. 48–49)

The strength of the argument lies in its rejection of the null hypothesis. As Ridley explains, there would be no such similarities across the species under the "separate creation" model. Similarly, Jerry Coyne repeats this religious premise that under the creation model, the anatomies of different species should share no common patterns but rather should be unique:

> There is no reason why a celestial designer, fashioning organisms from scratch like an architect designs buildings, should make new species by remodeling the features of existing ones. Each species could be constructed from the ground up. (Coyne 2009, p. 54)

Likewise, the fossil record should be random rather than exhibiting patterns: "So the appearance of species through time, as seen in the fossils, is far from random" (Coyne 2009, p. 29). These are longstanding, powerful arguments for evolution, based on observed, non-random patterns in nature.

Therefore, while the testing of evolutionary models against random chance models superficially appears to be nothing more than a false dichotomy, such arguments have traditionally been cast within a much broader religious context. That is, the test is not comparing two narrowly construed hypotheses, but rather the test is comparing the creation versus evolution narratives, broadly defined as the two possible explanations. Therefore, the null and alternative hypotheses are logically complementary and the argument is valid. The underlying religion supplies the necessary premise about how a creator would create the world; namely, a random design, thus making it possible to subject the creation model to such a test.

## 6. Conclusions

There is a long history of statistical hypothesis testing in origins studies. This history traces back to the early eighteenth century, long before the formalization of modern statistical methods and their nomenclature, such as NHST and *p*-values. Across this history of hypothesis testing in origins studies, there is a remarkable consistency in the assumptions, methods, and claims made—this in spite of a three-century time span involving thinkers from various lands, traditions, and scientific disciplines. One important consistent theme in this history of hypothesis testing is that the null and alternative hypotheses, random and non-random designs, respectively, are not well understood. In fact, it is an open question as to whether either are even possible, and even if they are, there is no mathematical or scientific reason to think the two hypotheses are logically complementary. This appears to invalidate the standard technique of accepting the alternative hypothesis upon rejection of the null hypothesis.

In other words, from a statistical perspective, this centuries-long tradition of applying NHST to origins studies is built on a fallacy, namely, "the natural world is non-random therefore it must have evolved". This would be a remarkable finding given the strength of the triumphant claims made, the stature of the practitioners involved, and the influence of their findings. But did all of these practitioners really make such a naïve blunder?

The answer is no. This criticism, from a statistical methods perspective, does not account for the full breadth of the claims being made. For these claims are made from within a broader historical context. The triumphant claims made with conviction are not the result of a repeated and naïve logical fallacy, but rather a larger religious framework regarding the creator and creation. In this framework, the world must be created in a random manner consistent with a full creation that realizes every possibility (the potential parallel with plenitude thinking is beyond the scope of this paper). Therefore, the logic becomes, "creation is non-random therefore it must not have been intentionally designed, and instead must have evolved". Sometimes this underlying religious assumption is explicitly stated. Other times it is left unspoken. But it is always crucial. Without this religious assumption the hypothesis testing and its results are, in fact, fallacious and non-sensical.

What is the impact of this religious assumption? As we saw above, NHST has been roundly criticized. One of those criticisms is that it has harmed science. In origins studies that also is true. Most obviously, a religious premise has underwritten the view that a random pattern represents the creation model. It would be difficult to imagine a more suspect design model of the natural world. This is highlighted by Kant's suggestion that the solar system should be so much random motion, or Darwin's suggestion that "the bones of the arm and forearm, or of the thigh and leg" should be reversed sometimes, as though such wholesale restructuring would have no functional effects. Today's assumption that all of biology would be randomized if it were created, as exemplified in the more recent studies, is no less suspect.

Furthermore, this philosophy of science obviates mechanism. The motivation is from pattern, not mechanism. These theories for the origin of cosmology and biology were not motivated by the finding of a compelling, explanatory mechanism, by which the world could have arisen, but rather by

the patterns found in nature. The alternative hypothesis can be supported even though it lacks specific, detailed, or even realistic mechanisms. The development and testing of the scientific mechanisms can come later and can sustain many failures. Indeed, mechanisms that are proposed typically do not even explain the very data used to reject the null hypothesis. Variations in the planetary inclination angles, or variations in the species hierarchical structure, are not explained by the condensing nebula or common ancestry hypotheses, respectively. Additional, ad hoc, contingent events and causes are required.

But in spite of this lack of mechanism and shortcomings of the alternative hypothesis, it is assumed to be true because the *p*-value is so small. This philosophy of science, with its statistically powerful rejection of the null hypothesis, leads to a radically high, unjustified confidence in the alternative hypothesis. This, in turn, undercuts the scientific process, for it relieves the alternative hypothesis of healthy scrutiny. There is a lowering of the bar and significant problems can be overlooked or addressed with speculation. Bernoulli, Buffon, Kant, and Laplace could proclaim full confidence in their respective theories with no demonstration that their mechanisms actually could, in fact, create the solar system. The same dynamic has played out in biological evolutionary theory. Shortcomings in proposed mechanisms are viewed merely as interesting research problems rather than evidence against the theory. This makes the evolutionary theories resistant to rigorous testing.

All of this presents a new understanding of the influence religion can have on science. Specifically, in this case study of hypothesis testing in origins studies, we see religious assumptions governing and directing scientific conclusions. This suggests a science–religion interaction in which science is subservient to religion. Rather than the warfare thesis, which views science and religion in conflict, or the cooperation model, which sees science and religion working together to some extent, here we see religion dictating certain doctrines to be incorporated and affirmed by science. This influence should be considered in models of the interaction between science and religion.

**Funding:** This research received no external funding.

**Conflicts of Interest:** The author declares no conflict of interest.

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
