# Peer review of "On the Influence of Religious Assumptions in Statistical Methods Used in Science"

_religions, doi:10.3390/rel11120656_

Round 1

Reviewer 1 Report

The paper is well written and the subject interesting; but I have a number of observations.

Meanwhile, the title. The term statistical test leads the reader to think the author wants to perform a statistical analysis or in any case report numerical statistical analyzes by others. That is not the case. Even the term religious seems to me not very suitable. In the examples that the author presents, more than a structured religion, reference is simply made to the belief in a demiurge. Probably a more suitable title would be Influence of religious beliefs in probabilistic considerations on the history of science.

The lengthy considerations of the introduction on the p-value seem to me substantially useless. Among other things, it is not even clearly explained what p-test is, and talking about numbers, such as p = 0.01 or p = 0.05, as done in chapter 2 seems to me misleading and useless. Possibly the first rows of chapter 2, 56-78, should be erased.

Chapters 3 and 4 are too long compared to what the title provides.

Chapter 5, of some interest, clearly shows what the author means by statistical analysis; it is simply an intuitive reasoning on the likelihood or plausibility of a hypothesis; there is no probability calculation or statistical analysis (even though some numerical reference is reported in chapters 3 and 4).

What the author says about logical complementarity is not clear to me (see rows 289-290).  Consider for instance the two hypotheses: A) there is no common ancestor. B) there is at least one common ancestor, they seem logically complementary to me. I didn't quite understand thus what the author means.

Basically, in order for the article to be published in my opinion, it is necessary that the author:

would change the title

would do away with the statistical terminology; in case he could keep the concept of NHST (weakened) but no that of p-test

should possibly reduce chapters 3 and 4 and lengthen 5, alternatively he has to change the title completely, putting religion into the background; but it would be a pity because the considerations about religious beliefs are interesting.

Author Response

Please see attached file, author responses are highlighted in red.

Reviewer 2 Report

The author presents an argument against the use of statistical testing to support evolutionary hypotheses. If correct, this would be really surprising since this kind of argument is very common among evolutionary biologists. However, I do not think that the argument in this paper succeeds. 

The author argues that in the case of hypothesis testing in origin studies, the two competing hypotheses are not logically complementary. I do not see why this is true. In a case of a set of species, for example, there are two logically complementary possibilities: either they share a common ancestor, or they do not. The author suggests that the reason these two hypotheses are not logically complementary is that ‘the two hypotheses are not well understood’ (133) and ‘researchers do not know how the evolution of the species occur’ (135).  However, I do not see why this is relevant. The fact that we may not know how exactly evolution occurs, does not mean that we cannot infer common ancestry given a specific pattern among species. The author writes that ‘we do not even know the set of all possible ways that the species could have arisen’ (149-50). It is true that there are many ways species could have evolved and that we have yet much to learn about evolutionary mechanisms and about the exact role of natural selection in a given case of speciation. But, the question whether evolution has occurred and what exactly are the mechanisms of evolution, are two separate issues. One can have strong evidence in favour of the former but not know much about the latter. This is, for example, what happened in the end of the 19th century; biologists were convinced that evolution has occurred and that species share common ancestors, but there was much debate about the exact mechanisms of evolution.

The author seems to understand the hypothesis of common ancestry as involving both the claim that species, for example, share a common ancestor, plus a specific claim about how evolution actually occurred. It is true that in this sense, there are many different hypotheses of evolution, each involving a different combination of mechanisms. However, this is not how the argument that the author examines is typically construed. Specifically, this is not how Darwin construes the argument; Darwin compares the hypothesis that species were created independently, with the hypothesis of common ancestry (and not with a specific hypothesis about evolution that includes evolutionary mechanisms).

In section 5 the author claims that the null and the alternative hypotheses are logically complementary after all, since ‘the underlying religion supplies the necessary premise about how a creator would create the world’ (351-52). The thought here is that ‘if the world was created, then it would manifest random patterns. When such patterns are not observed, they would therefore be concluded to have arisen by some sort of evolutionary process’ (291-93). I am not sure how a religious premise can make the argument valid, as the author thinks. Even if we were to accept what the author says about the false dichotomy, how can the religious context alter this situation? If the author thinks that in a religious context the dichotomy is creation vs evolution in a broad sense (i.e. not including any specific evolutionary mechanisms), then why cannot we construe the argument outside a religious context in the same way (thereby making the two hypotheses logically complimentary as explained above)?

Author Response

(The authors gave the same response as above.)

Reviewer 3 Report

I have included everything that is unclear or that should be improved in the attached paper. I have been dealing with hypothesis testing all my life, but the author(s) bring up some new perspective. However, they have to do the suggested modifications.

Best regards

Author Response

Handwritten comments incorporated into paper, please see revised manuscript.

Round 2

Reviewer 1 Report

Reviewer 1

In green my reply.

The paper is well written and the subject interesting; but I have a number of observations.

Meanwhile, the title. The term statistical test leads the reader to think the author wants to perform a statistical analysis or in any case report numerical statistical analyzes by others. That is not the case.

Change made to clarify this, similar to title supplied at end of this paragraph.

Even the term religious seems to me not very suitable. In the examples that the author presents, more than a structured religion, reference is simply made to the belief in a demiurge.

Difficult to respond to because there is no reference to a demiurge. Section 5 gives examples from Kant referring to “God,” Darwin referring to “the Creator,” Ridley referring to the “creation model,” and Coyne referring to a “designer.” These are all clearly religious claims.

I do not agree with the author. It is probably a question of words, but with religion usually one considers a structured religion, that is Christianity, Hinduism, Islam and so on.  The simply belief in the  existence of a creator (or a demiurge, it’s the same), in my opinion is more a question of metaphysics than religion.

Probably a more suitable title would be Influence of religious beliefs in probabilistic considerations on the history of science.

Yes, done.

The lengthy considerations of the introduction on the p-value seem to me substantially useless. Among other things, it is not even clearly explained what p-test is,

I have clarified this and added an example.

The definition of p-test of the author is unintelligible; please verify. Moreover I cannot see the example.

and talking about numbers, such as p = 0.01 or p = 0.05, as done in chapter 2 seems to me misleading and useless. Possibly the first rows of chapter 2, 56-78, should be erased.

Difficult to respond to because these probabilities are discussed after Line 78, and they help to contextualize the discussion, and theme of the paper.

I strongly advise the author to erase lines from 56-to 78; of corse by modifying the text to account this cut.

It seems to me that  the author has not stressed the difference between statistics and probability. Probability is an axiomatic mathematical theory; statistics concerns an experimental activity generally devoted to estimate parameters of a given probabilistic model (mean value, variance and so on).

While the studies on biology and on origin referred to in chapter 3 (though there are no details) seem to be statistical in nature, the case referred to by Bernoulli and Buffon in chapter 4 are simply probability evaluations. How can the author consider them as statistical analyses?

Chapters 3 and 4 are too long compared to what the title provides.

The title is now changed.

The problem remains

Chapter 5, of some interest, clearly shows what the author means by statistical analysis; it is simply an intuitive reasoning on the likelihood or plausibility of a hypothesis; there is no probability calculation or statistical analysis (even though some numerical reference is reported in chapters 3 and 4).,

No this is not merely intuitive reasoning. As noted, detailed numerical results are given in Sections 3 and 4, so there is no need to repeat them here in Section 5. The point of Section 5 is explain the history of the non empirical, religious claims underwriting the calculations.

First I suggest to change the title of section 5, as it concerns not only origin but also cosmological questions.

I do not agree with the author. I repeat there are problems, in my opinion, in the use of the concept of statistics. In section 4 there are no statistical analysis; moreover the reader is surely curious to see as Kant dealt with probabilistic (or statistic?) matters in section 5; I mean how he obtained numerical values for probability.

What the author says about logical complementarity is not clear to me (see rows 289-290). Consider for instance the two hypotheses: A) there is no common ancestor. B) there is at least one common ancestor, they seem logically complementary to me. I didn't quite understand thus what the author means.

Yes, (A) and (B) are logically complementary. But as Section 3 explains, (A) is modeled as random species, such that any non randomness is taken as proof of a common ancestor. I have added more clarification at Line 153.

It is a little better, but not completely clear.

Basically, in order for the article to be published in my opinion, it is necessary that the author:

would change the title

Done, as recommended above.

would do away with the statistical terminology; in case he could keep the concept of NHST (weakened) but no that of p-test

As explained above, the statistical terminology, such as null hypothesis and p-value, are key concepts, at the core of the evolutionary studies which make up the bulk of the paper.

Yes I know, but I do not agree the way you use statistical concepts.

should possibly reduce chapters 3 and 4 and lengthen 5, alternatively he has to change the title completely, putting religion into the background; but it would be a pity because the considerations about religious beliefs are interesting.

Done. Title changed as recommended.

Reviewer 2 Report

The author advances two main claims: (i) that in the case of hypothesis testing in origin studies, the two competing hypotheses are not logically complementary, which makes the argument faulty; (ii) that  the addition of a religious premise makes the argument valid. I do not think that the author has done enough in the revised version of the paper to make a convincing case for claim (i). 

The author mentions various considerations that he/she takes to show a fundamental difference between NHST as applied to the case of evolution and as applied to the test of a drug. In his reply, the author mentions that these considerations are not meant to show the non-complementarity of the two competing hypotheses in the case of common ancestry testing -but then, what is the role of these considerations? (lines 116-139).

In his reply the author states: ‘A non random data set does not force one to adopt the common descent hypothesis, for it is not known that common ancestry is the only alternative to random species’. This does not seem very illuminating: the two hypotheses are not ‘a random looking pattern among species’ vs ‘a non-random pattern’; instead, the two hypotheses are that the species do not share a common ancestor vs that they do share one, which seem to be logically complementary. Again, the argument against claim (i) needs to be clarified in the paper.

Round 3

Reviewer 2 Report

In his response to the problem that the two hypotheses seem logically complementary, the author states:

To summarize, the reviewer interpreted the common descent case as follows: 

Null hypothesis: the species do not share a common ancestor 

Alternative hypothesis: the species share a common ancestor 

This is not the test in question. As I clarified in Lines 113-4, the test in question is between “random looking pattern” and “common descent.” This is the test in the given example, and this is the test in the studies cited later in the paper. It is not between “the species do not share a common ancestor” versus that “they do share one.” That is not the test that is being run. Lines 113-4 clarify this. But to explain this further, I have now added more clarification at Lines 154-155, as well as clarifications at Lines 20-25’. 

While it is clear in the paper that this is how the author interprets the test, this is not how the test is described in the studies mentioned by the author. For example, in the White et al study mentioned in the paper, the authors state: ‘Our null model can be considered in the following way - that the taxa in subgroup X are descended from an unknown number 1< = rX< = |X| of root sequences, the taxa in subgroup Y are descended from an unknown number 1< = rY< = |Y| of root sequences, and that the rX+rY root sequences are all independent from each other’. Clearly here, the null hypothesis is independent ancestry. Similarly, in the Baum et al study also mentioned in the paper, the alternative to common ancestry is taken to be ‘separate ancestry’. The hypothesis of ‘random design’ is not directly compared to the hypothesis of common ancestry in these studies. ‘Random-looking patterns’ is used in the Baum et al study to predict distributions given the null hypothesis. So, Baum et al state: ‘A null distribution under species SA was obtained by randomly permuting character states among taxa’. This last move is probably why the author interprets the test as comparing the two hypotheses of ‘random design’ and common descent. However, this has to be explained in the text, and it has also to be explained how exactly, given all this, the non-complementarity arises.

Author Response

Reviewer 3

While it is clear in the paper that this is how the author interprets the test, this is not how the test is described in the studies mentioned by the author. For example, in the White et al study mentioned in the paper, the authors state: ‘Our null model can be considered in the following way - that the taxa in subgroup X are descended from an unknown number 1< = rX< = |X| of root sequences, the taxa in subgroup Y are descended from an unknown number 1< = rY< = |Y| of root sequences, and that the rX+rY root sequences are all independent from each other’. Clearly here, the null hypothesis is independent ancestry. Similarly, in the Baum et al study also mentioned in the paper, the alternative to common ancestry is taken to be ‘separate ancestry’. The hypothesis of ‘random design’ is not directly compared to the hypothesis of common ancestry in these studies. ‘Random-looking patterns’ is used in the Baum et al study to predict distributions given the null hypothesis. So, Baum et al state: ‘A null distribution under species SA was obtained by randomly permuting character states among taxa’. This last move is probably why the author interprets the test as comparing the two hypotheses of ‘random design’ and common descent. However, this has to be explained in the text, and it has also to be explained how exactly, given all this, the non-complementarity arises.

Submission Date

16 October 2020

Date of this review

28 Nov 2020 11:04:42

Author’s response:

The reviewer insightfully notes examples of the disparity between (i) the statistical test that is performed and (ii) how that test is described. The test is between random design and common ancestry, but it is described as between (i) separate ancestry and (ii) common ancestry. The strength of the test relies on the use of random design in the test. This is crucial. In all of the historical studies discussed in the paper, going back to the eighteenth century, the authors always use random design, with an attached metaphysical interpretation. The reviewer writes: “this has to be explained in the text.”

The Conclusion section (Section 6) now does explain this as follows:

=====

In this framework, the world must be created in a random manner consistent with a full creation that realizes every possibility (the potential parallel with plenitude thinking is beyond the scope of this paper). Therefore, the logic becomes, “creation is non-random therefore it must not have been intentionally designed, and instead must have evolved.”

=====

Also, the final paragraph in Section 5 now fleshes this out. The reviewer also writes: “and it has also to be explained how exactly, given all this, the non-complementarity arises.”

Well recall that the paper argues that when the tests are properly understood, there is no non-complementarity. The non-complementarity arises when the tests are interpreted literally, outside of their historical context. This apparent non-complementarity is explained now in several places within the paper, such as in Section 3, which states:

=====

In the evolution example, it is not known that common ancestry is the only alternative to random species. This means that there could be other explanations for the observed patterns amongst the species. In fact, a rejection of the null hypothesis simply means that the observed data are not random. Otherwise it says little about the actual structure of the data, and how well (or badly) they fit the evolutionary common descent model. This is an important distinction. In NHST, the null and alternative hypotheses must be carefully designed such that they are logically complementary. Therefore, if the null hypothesis is false, then the alternative hypothesis must be true, by definition. But in the evolution example this does not hold. Not only does the failure of the null hypothesis say nothing about the veracity of the alternative hypothesis, but a small p-value can be obtained even if the data have a low probability on the alternative hypothesis. In other words, simply put, the data might not fit both the null and alternative hypotheses very well. … This evolution example reflects actual studies that have been published. As described above the studies use a random pattern null hypothesis test, show that the observed species data do not fit the model well, reject the null hypothesis, and argue forcefully for evolution, the alternative hypothesis.

=====

So I believe the paper now does provide the explanations suggested by the reviewer.